# Prolonged Visual Evoked Potential Latencies in Dogs Naturally Infected with Canine Distemper Virus

**DOI:** 10.3390/v16111721

**Published:** 2024-10-31

**Authors:** Mary Gutiérrez, Luis Delucchi, Alejandro Bielli, José Manuel Verdes

**Affiliations:** 1Small Animals Medicine Unit, Department of Clinics & Veterinary Hospital, Clinical Neurology, Faculty of Veterinary, Universidad de la República (Udelar), Route 8 Km 18, Montevideo 13000, Uruguay; bobby95ld@gmail.com; 2Histology & Embriology Unit, Department of Veterinary Biosciences, Faculty of Veterinary, Udelar, Montevideo 13000, Uruguay; biellialejandro@gmail.com; 3Pathology Unit, Department of Pathobiology, Faculty of Veterinary, Udelar, Montevideo 13000, Uruguay; jose.verdes@fvet.edu.uy

**Keywords:** canine distemper virus, dogs, electrophysiology, increased latency, veterinary neurology, visual evoked potentials

## Abstract

Canine distemper (CD) is a deadly, multi-system infection caused by a Morbillivirus. The canine distemper virus (CDV) frequently affects the nervous system with demyelinating leukoencephalitis, the most common neurological lesion. The disease has been linked to multiple sclerosis (MS) in humans due to similar clinical presentation and pathophysiological mechanisms. In MS, visual evoked potentials (VEPs) have been identified as a reliable marker for disease progression, enabling the early detection of clinically suspected lesions. The aim of this study was to determine if there are any abnormalities in VEP responses in dogs with neurological CD. Visual evoked potentials and electroretinogram (ERG) were recorded at both the cranial and spinal levels in dogs naturally infected with CDV and in healthy dogs. The results in the CDV-infected group revealed a bilateral increase in the latency of N1, P1, N2, P2, and N3 waves of the VEPs, without any alterations in their amplitudes. No significant differences were observed in the ERG between the groups. These results suggest that altered VEP responses could serve as an early diagnostic indicator of neurological damage caused by distemper. Therefore, conducting these studies could potentially aid in the detection of central nervous conduction disorders during the subclinical phases of the disease.

## 1. Introduction

Canine distemper virus (CDV) is the etiological agent of canine distemper (CD), one of the most common infectious multisystem diseases in domestic dogs and wildlife. It also affects other members of the orders Carnivora, Rodentia, Primata, Artiodactyla, and Proboscidea, being a potential threat to wild and endangered species [1].

It spreads through the respiratory tract and induces immunosuppression, which favors secondary infections [2,3,4]. Despite the fact that this disease has reduced in prevalence, due to the use of vaccines, numerous infectious outbreaks have been described in various parts of the world [5]. Although the disease has been known for many years, there is still no effective treatment, which has led CD to become an endemic disease throughout the world [6].

Some of the neurological signs that can occur in cases of CD are neck stiffness, vestibular and cerebellar signs such as ataxia and nystagmus, paresis, paraplegia, seizures, and myoclonus, with the latter being highly suggestive of the disease. Visual deficits such as chorioretinitis and optic neuritis have also been described [7,8,9,10,11].

CD has been proposed as a model for different demyelinating conditions in humans [12]. In particular, CD shares several neuropathological similarities with multiple sclerosis (MS) [2,13]. Similarly, Amude et al. [7,8] propose neurological CD as “acute MS in dogs.” Some of the shared clinical signs of both diseases are optic neuritis, retinal alterations, paresis, motor incoordination, instability, and involuntary movements such as myoclonus [14,15,16,17].

In MS, evoked potentials (EPs) have been proposed as solid biomarkers, since abnormalities on them can be found in the early stages of the disease, prior to the manifestation of clinical signs and functional impairment. Multimodal EPs have been shown to be highly correlated with, and useful predictors of, the disability that the disease will cause [18,19,20].

These techniques allow for the early detection of functional alterations prior to their clinical manifestation, even in the absence of deficit signs [20,21,22,23,24].

In addition, EPs can help to differentiate between demyelination and axonal damage [15,25,26,27].

One of the most commonly employed EPs in MS diagnosis are visual evoked potentials, and they have been shown to be useful to detected MS-associated optic neuritis, which occurs in more than half of MS patients.

VEPs are a commonly used method for the electrophysiological diagnosis of MS-associated optic neuritis [28,29,30]. More than 50% of patients develop optic neuritis (ON) in MS. ON occurs as the first manifestation in 20 to 30% of cases [14,17,31,32]. In addition to this, it has been demonstrated that MS can cause direct damage to the retina, including the thinning of the inner and outer nuclear layers, as well as alterations in the retinal nerve fiber layer and the ganglion cell layer. The function of the retina can be evaluated by means of another neurophysiological technique, the electroretinogram (ERG).

To date, there are few studies that describe how VEPs and ERG are affected in dogs with CDV infection.

Ochikubo et al. [33] studied visual evoked potentials in squirrel monkeys infected with DCV, finding alterations in the subacute stage of the disease, as well as finding alterations in an animal without clinical signs, but which showed the histopathological changes typical of the disease at autopsy. Richards et al. [34] reported a case of a 9-year-old Jack Russell terrier dog with seizures, circling, and blindness, who underwent ERG without any alterations being found.

Consequently, the present study proposes to study the damage that this neurodegenerative disease causes naturally in domestic dogs, using neurophysiological techniques, and to evaluate whether there is functional impairment of the visual pathways in dogs naturally infected with CDV using both VEPs and ERG.

## 2. Materials and Methods

### 2.1. Study Population

For the present study, two groups of dogs were established, and no distinction between sex, breed, and vaccinal status was made. Group 1 (control) consisted of ten healthy animals that did not show any signs of general disease and had a normal clinical examination. Group 2 (clinical cases) consisted of thirty-five animals with a confirmed diagnosis of CD (for details of the age and clinical signs of each CDV-infected dog of Group 2, see Appendix A). The diagnosis was based on clinical signs and the qualitative detection of antigens of the canine distemper virus (CDV) using an analytical kit, in ocular, nasal, and urine discharge (Fast test Distemper^®^, Megacor, Bregenz, Austria). A real-time RT-PCR assay was also performed for the detection and quantitation of CDV. Animals were excluded if the presence of myoclonus was either so extensive or intense that the muscle contraction induced artifacts on the EPs, making their interpretation unreliable.

Visual evoked potentials and ERGs were performed on every dog. In the animals whose clinical condition allowed it, the studies were carried out under sedation with Xylazine (Xilased^®^, Vetcross, Buenos Aires, Argentina) (0.5–1.0 mg/kg) per intramuscular via (I/M) [35,36,37].

### 2.2. Evoked Potential Recordings

The bioelectric signals were recorded with a digitized signal amplification, averaging, and filtering system (Sistema Bio-PC Potentiales Evocados^®^ V.9, Akonic S.A., Buenos Aires, Argentina). Subdermal stainless steel needle electrodes were used, including one active or recording (+) close to the generating area of the electrical responses, another reference (−) far from the previous one, and a third ground electrode. For VEPs, the recording electrode was placed on the midline of the occipital protuberance (Oz), the reference electrode on the midline of the frontal bone between both eyes (Fpz), and the ground electrode on the midline of the vertex (Cz) between both ears [38,39,40]. For the ERG, the same type of electrode was used, wherein the recording electrode was placed half a centimeter below the lower eyelid, while the reference electrode was placed half a centimeter caudal to the lateral edge of the eye to be stimulated. The ground electrode was placed on the midline of the vertex (Cz).

As stimuli to evoke the responses, flashes of white light generated with a stroboscope at a frequency of 1 Hz were used, placed close to the eye to be stimulated without touching the eyelids. A total of 128 monocular stimulations were performed for the VEPs and 64 for the ERGs, covering the contralateral eye with an opaque patch to isolate the responses of each eye and, therefore, to detect functional asymmetries between both eyes. For each technique, two complete sets of stimuli were performed on each eye. The two consecutive recordings in each eye confirmed the reproducibility of the responses. The signal was filtered using a 1–100 Hz passband, with an analysis time of 300 milliseconds. The impedance for all electrodes was checked to be less than 5 kOhm.

### 2.3. Statistical Methods

Latency (time elapsed between the application of the stimulus and the maximum peak of the positive or negative wave, in ms) and amplitude (measured from the maximum positive or negative value at the peak of the previous or subsequent wave, in microvolts -µV-) values were evaluated. Normality was evaluated by means of the Shapiro–Wilk test, and, since the data did not show a normal distribution, the differences between the groups were evaluated using the non-parametric Mann–Whitney–Wilcoxon U test for independent samples. Values were considered significant at *p* ≤ 0.05.

## 3. Results

### 3.1. Population

The dogs’ ages ranged between 3 and 144 months, with an average of 22.78 months.

### 3.2. Visual Evoked Potentials

In the group of healthy animals (Group 1) the flash VEPs consisted, according to their polarity, of three negative and two positive components, recorded in the first 150–200 ms after visual stimulation. Figure 1 shows the records of both a healthy animal and an animal infected with VDC.

In most of the waves of VEPs performed on Group 2 (*n* = 35), a significant increase was observed bilaterally in all latencies of potential. Table 1 shows the mean latencies of each wave for each group. In one animal, no response was obtained bilaterally, while, in four dogs, the responses were only obtained unilaterally.

The latency of the N1 component was higher in the dogs with DC by 76.5% in left eye and by 64.7% in the right eye. In four of the animals, no response was obtained unilaterally. For P1, an increase in latency was found in 76.5% of the animals in the left eye and 70.6% in the right eye. Four of the animals did not respond unilaterally. N2 latency in the left eye was prolonged in 70.6% of cases, while the values for the right eye were prolonged in 58.8%. Six of the dogs with CD did not present this wave bilaterally, and one dog presented it only in the left eye. The P2 component had an increase in latency in 32.3% of the animals in the left eye and in 35.3% of the animals in the right eye. Five of the dogs did not present this wave bilaterally, while, in two dogs, the latency increase was unilateral. Regarding the N3 wave, consistent and reproducible records were only obtained in nine of the thirty-five animals studied (26.4%). A bilateral increase in latency was found in 11.7% of the animals.

No differences in amplitudes were found between the groups; however, the N2-P2 amplitude tended to be higher in the group of CD animals for the left eye, with the mean and standard deviation of the controls being 1.02 ± 0.58 and 1.66 ± 1.06 that of the affected animals, respectively (*p* = 0.08).

### 3.3. Electroretinograms

In all of the animals, ERG by flash was characterized by the presence of a positive wave (wave a) followed by a wave of great negative amplitude (wave b), without the presence of wave c. The latencies corresponding to waves a and b in the ERG were not different between the groups. Likewise, the amplitudes were not affected.

Of the eighteen CD-infected animals that underwent ERG, six presented altered VEPs (five with increased latencies and one with no response).

## 4. Discussion

Considering that there is no consensual nomenclature of the waves, in this study, it was decided to name them descriptively according to their polarity, taking into account the inputs of the differential amplifier and how the polarity was determined according to it [41].

The development of this study provides new data, which can be used both in the diagnosis and in the follow-up of patients with CD, as well as for other demyelinating, neurological, and ophthalmological diseases.

Although the use of evoked potentials in dogs has been primarily focused on research rather than on clinical applications, the normal response of VEPs and ERGs in dogs has long been described by several authors [39,41,42].

In the present study, no distinction was made between sex, age, and vaccinal status. There are no reports that justify making a distinction between sexes; moreover, human studies have indicated that differences in VEP values in men versus women were not clinically significant [29,30].

Regarding the changes that occurred in VEPs in relation to age, Kimotsuki et al. [43] report a prolongation in the latencies of the N2, P2, and N3 waves (adequate nomenclature according to the polarity used in this study) in the VEPs of dogs older than 15 years when compared with the VEPs of 1-year-old animals. On the other hand, there are reports that mention that the maturation of the visual pathway in dogs occurs between 11 and 15 weeks, when amplitudes increase and latencies decrease [40,44].

In this study, VEPs are altered in the animals infected with DC, as compared to the control group. According to the hypothesis, the latencies in the CD group were prolonged VEPs. These findings are consistent with reports regarding MS, given the neuropathological similarities between both diseases [18,45].

Regarding the alterations found in the VEPs performed on dogs with CD, the latencies of all components were increased, while the amplitudes did not undergo changes, except for N2-P2 for the left eye.

On the other hand, in MS, it has been shown that an increase in the latency of the VEP waves occurs (performed by pattern), with an increase in P100 latency being the main alteration described. This has been associated with disorders of myelination [46]. A demyelinating lesion in the optic nerve of approximately 10 mm in length causes a conduction delay of approximately 25 ms [19]. MRI studies in patients with MS have shown lesions in optical radiation in 70% of individuals [47], which explains the visual dysfunction present in them.

In our study group, it was observed that the increases in latencies were more marked in the P1, N1, and P2 waves. This increase in latency is consistent with what is expected in connection with the pathophysiology of DC.

Infection with DCV leads to lesions in the myelin sheaths and the formation of vacuoles in the white matter [48]. With regard to neuropathological lesions that could influence the functioning of the visual pathway, Amude et al. [7,8] described that the occipital cortex is mainly affected in old dog encephalitis. Mike and Carithes [49] observed histopathological lesions in the eye of infected dogs, as well as signs of demyelination of the optic nerve and its tracts. This set of alterations is in agreement with our findings, since the prolongation of VEP latency has long been considered an indicator of demyelination. Moreover, the degree of demyelination within the visual pathway of animal models correlates with the magnitude of delay [26].

Regarding the increase in the N2-P2 amplitude in the VEPs for the left eye, these results were not expected, given that, in the reports of EP alterations due to demyelinating lesions with nerve conduction involvement, an increase in latency, and a decrease in amplitude usually occur.

The increased amplitude of the VEPs, seen in our animals infected with CDV, could only be observed on the left side, which can be explained by having always started the stimulations in the same eye (right), while the other remained occluded. This allows for an adaptation to darkness in the second eye, with the consequent increase in amplitude.

Regarding the ERG results, Mike and Carithes [49] found in dogs infected with CDV lesions at the histopathological level of the following types: necrosis, degeneration of the retinal ganglion cells, edema and peri-vascular cuffings, proliferation pigment epithelial cells, retinal inclusion bodies, and rod and cone atrophy. However, in our study, the absence of specific alterations in the electroretinography in the diseased animals agrees with what has been demonstrated for this technique in patients with MS, given its low specificity [18].

Klistoner et al. [26] found alterations in patients with MS, which are presumably due to the thinning of the retinal ganglion cells. However, these lesions are more evident by optical coherence tomography. Given its low specificity and the additional time involved in performing the electroretinogram, it would not provide any clinical benefit in the routine management of CD.

## 5. Conclusions

The existence of alterations in the VEPs (prolonged latencies) was verified. The increase in VEP latencies indicates that neurological lesions can be detected before neurological signs appear. These techniques have been found to be a very useful tool in the detection of silent lesions, since signs related to nervous system involvement in patients with CD often appear later than signs in other organ systems.

Thus, we propose VEPs as a good tool to evaluate the effectiveness of different treatments against CD in future studies.

## Figures and Tables

**Figure 1 viruses-16-01721-f001:**
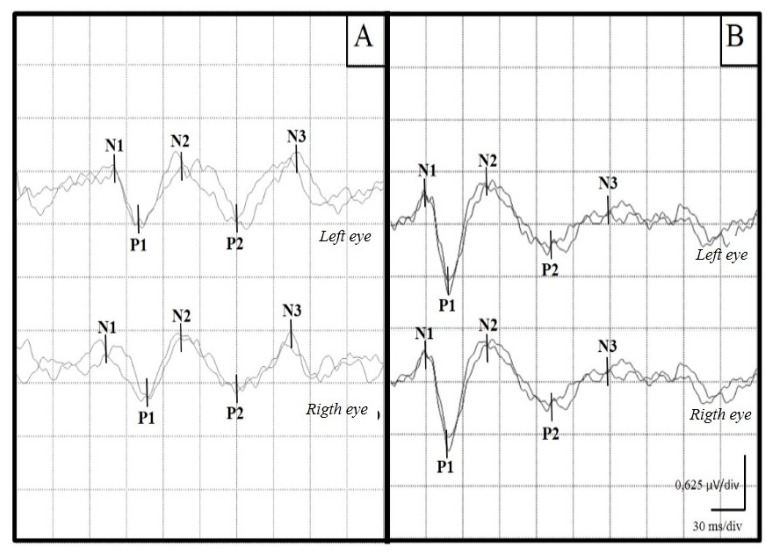
Representative examples of VEP recordings from a dog with CD (**A**) and a healthy control dog (**B**). An increase in the latencies of the waves can be seen in the sick canine. The negative waves are named as N, the positive waves are named as P. In both wave types, numbers 1, 2 or 3 means temporal order of appearance of each wave.

**Table 1 viruses-16-01721-t001:** Mean latencies and standard deviations for each component of the VEPs in both groups. Values (mean ± standard deviation) marked with different letters within the same row are significantly different (*p* < 0.05).

	Distemper		Control	
	Left Eyes	Right Eyes	Left Eyes	Right Eyes
N1	56.5 ± 21.0 ^a^	54.9 ± 21.9 ^a^	27.6 ± 3.1 ^b^	27.4 ± 3.1 ^b^
P1	79.5 ± 25.5 ^a^	76.1 ± 26.0 ^a^	45.3 ± 4.4 ^b^	45.0 ± 3.6 ^b^
N2	107.0 ± 26.2 ^a^	101.0 ± 29.3 ^a^	67.4 ± 7.0 ^b^	70.0 ± 11.2 ^b^
P2	129.3 ± 39.7 ^a^	123.3 ± 35.1 ^a^	96.5 ± 11.0 ^b^	95.8 ± 14.9 ^b^
N3	159.9 ± 31.7 ^a^	171.7 ± 33.7 ^a^	127.0 ± 6.7 ^b^	125.4 ± 20.8 ^b^

## Data Availability

All the data are included in the manuscript.

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
