# Peer review of "Prolonged Visual Evoked Potential Latencies in Dogs Naturally Infected with Canine Distemper Virus"

_viruses, 2024, doi:10.3390/v16111721_

Round 1

Reviewer 1 Report

Comments and Suggestions for Authors

The manuscript "Prolonged visual evoked potential latencies in dogs naturally infected with Canine Distemper Virus" aims to evaluate whether there is functional impairment of the visual pathways in naturally infected dogs by CDV using both VEP and ERG techniques. The article is well-written and it has scientific merit.

The methods are well described, but I suggest to add information about approval of animal research ethics committee (2.1). Other important point: briefly, explan how the VEP and ERGs findings are interpreted after the examination (2.2). 

Other minor detail:

Throughout the text - I suggest exchange "work" by "study", and "canine/s" by "dog/s"

Abstract

line 14, exchange "manifestation" by "lesion".

line 20-21, "The results revealed...their amplitudes." What group ?CDV-infected ? 

Introduction

line 75, exchange "turning in circles" by "circling"

Results

Table 1 - Include the meaning of LE, RE and add a, b,... after the values, when there is a statistical difference

Author Response

REVIEWER 1: Responses to comments and suggestions for Authors

The manuscript "Prolonged visual evoked potential latencies in dogs naturally infected with Canine Distemper Virus" aims to evaluate whether there is functional impairment of the visual pathways in naturally infected dogs by CDV using both VEP and ERG techniques. The article is well-written and it has scientific merit.

Comment 1: The methods are well described, but I suggest to add information about approval of animal research ethics committee (2.1).

Response 1 (2.1): We agree with this comment. Done.

Comment 2: Other important point: briefly, explain how the VEP and ERGs findings are interpreted after the examination (2.2).

Response 2 (2.2): Thank you for pointing this out. The values obtained for each individual were classified as normal or abnormal. The latency and amplitude thresholds to determine normalcy were based on the means and standard deviations of the control group data. A maximum latency and/or amplitude was considered abnormal if it exceeded two standard deviations. The alterations were classified as unilateral or bilateral depending on whether they affected one or both eyes. It was determined if there was a predominance of demyelinating phenomena, which could be verified through the delay of the potentials (increased latencies) or axonal damage proven by reduced amplitude (Klistorner et al. 2007; You et al. 2011).

Comment 3: Throughout the text - I suggest exchange "work" by "study", and "canine/s" by "dog/s"

Response 3: We agree with this comment. Done.

Abstract

Comment 4: line 14, exchange "manifestation" by "lesion".

Response 4: We agree with this comment. Done.

Comment 5: line 20-21, "The results revealed...their amplitudes." What group ?CDV-infected ?

Response 5: We agree with this comment. Fixed.

Introduction

Comment 6: line 75, exchange "turning in circles" by "circling"

Response 6: We agree with this comment. Done.

Results

Comment 7: Table 1 - Include the meaning of LE, RE and add a, b,... after the values, when there is a statistical difference

Response 7: We agree with this comment. Done.

Reviewer 2 Report

Comments and Suggestions for Authors

Dear Authors, the manuscript is oriented to study a relevant neurological aspect of CVD, but it is necessary Increase the information on the dogs recruited for group 2 - Clinical cases - to implement the scientific soundness  of the research study.

I suggest preparing a table that includes all 35 clinical cases associating the individual cases with the age of the animal and the neurological symptoms it may suffer.

On this basis, perform an appropriate statistical study such as Tamhane's T2 Test. It performs a comparison test of all pairs for normally distributed data with unequal variances.

Restructure the discussion and conclusions with the data returned by Tamhane's T2 Test.

Author Response

REVIEWER 2: Responses to comments and suggestions for Authors

Dear Authors, the manuscript is oriented to study a relevant neurological aspect of CVD, but it is necessary Increase the information on the dogs recruited for group 2 - Clinical cases - to implement the scientific soundness of the research study.

Comment 1: I suggest preparing a table that includes all 35 clinical cases associating the individual cases with the age of the animal and the neurological symptoms it may suffer.

Response 1: We agree with this comment. Done.

Comment 2: On this basis, perform an appropriate statistical study such as Tamhane's T2 Test. It performs a comparison test of all pairs for normally distributed data with unequal variances.

Response 2: Thank you for pointing this out. The data were statistically analyzed in accordance with the guidelines for the assessment of Evoked Potentials in humans published by the American Society of Clinical Neurophysiology. The recommended tests were used to create normality tables for evoked potentials in humans, as well as the use of two standard deviations to determine the degree of normality of the evaluated latencies and amplitudes. Recommended standards for normative studies of evoked potentials, statistical analysis of results, and criteria for clinically significant abnormalities; published on Journal of Clinical Neurophysiology, Volumen 23, Number 2, April 2006.

Comment 3: Restructure the discussion and conclusions with the data returned by Tamhane's T2 Test.

Response 3: Thank you for pointing this out. Please, see our comments in response 2.

Round 2

Reviewer 2 Report

Comments and Suggestions for Authors

Dear Authors, the manuscript is now fine and ready for publication.